# Efficient Design of a Clear Aligner Attachment to Induce Bodily Tooth Movement in Orthodontic Treatment Using Finite Element Analysis

**DOI:** 10.3390/ma14174926

**Published:** 2021-08-30

**Authors:** Kyungjae Hong, Won-Hyeon Kim, Emmanuel Eghan-Acquah, Jong-Ho Lee, Bu-Kyu Lee, Bongju Kim

**Affiliations:** 1Seethrough Tech. Corp., Seoul 06149, Korea; dent4u21@naver.com; 2Department Dentistry, College of Medicine, University of Ulsan, Seoul 05505, Korea; 3Dental Life Science Research Institute/Innovation Research & Support Center for Dental Science, Seoul National University Dental Hospital, Seoul 03080, Korea; wonhyun79@gmail.com; 4Department of Biomedical Engineering, Inje University, Gimhae-si 50834, Korea; eghanacquah@gmail.com; 5Department of Oral and Maxillofacial Surgery, School of Dentistry, Seoul National University, Seoul 03080, Korea; leejongh@snu.ac.kr; 6Asan Medical Center, Department of Oral and Maxillofacial Surgery, College of Medicine, University of Ulsan, Seoul 05505, Korea

**Keywords:** clear aligner, finite element analysis, attachments, orthodontics treatment, central incisor, stress distribution, overhanging attachment

## Abstract

Clear aligner technology has become the preferred choice of orthodontic treatment for malocclusions for most adult patients due to their esthetic appeal and comfortability. However, limitations exist for aligner technology, such as corrections involving complex force systems. Composite attachments on the tooth surface are intended to enable active control of tooth movements. However, unintended tooth movements still occur. In this study, we present an effective attachment design of an attachment that can efficiently induce tooth movement by comparing and analyzing the movement and rotation of teeth between a general attachment and an overhanging attachment. The 3D finite element modes were constructed from CBCT data and used to analyze the distal displacement of the central incisor using 0.5- and 0.75-mm-thick aligners without an attachment, and with general and overhanging attachments. The results show that the aligner with the overhanging attachment can effectively reduce crown tipping and prevent axial rotation for an intended distal displacement of the central incisor. In all models, an aligner with or without attachments was not capable of preventing the lingual inclination of the tooth.

## 1. Introduction

Traditionally, fixed appliances have been used in the orthodontic treatment of malocclusions. However, they are fast becoming the least patient-preferred choice for orthodontic treatment due to reported patient discomfort in chewing, maintaining oral hygiene [1,2,3], as well as the eventual wear of the enamel because of bonding agents used to attach the fixed appliances to the teeth [4,5]. Currently, orthodontic treatment using aligner technology is preferred to the traditional bracket and wire system due to their aesthetic appeal to patients and their relative comfortability [6,7,8,9,10]. Aligners have been demonstrated to be comfortable and reduce facial pain due to the relatively lower orthodontic loads delivered to the teeth as compared to the traditional fixed appliances [11]. In addition, aligners have been reported to minimize dental trauma, microbial risk, and apical resorption [12].

Although aligner technologies have improved significantly, there are still limitations in cases where corrections involve complex force systems. Tooth torque and rotation, as well as bodily movements, still pose a challenge with aligner technology [13]. The introduction of composite attachments has enabled the active control of tooth movement where frontal crowding can be corrected and bodily movements can be achieved [14,15].

Yokoi et al. [16] investigated the effects of attachments in closing diastema of maxillary dentition using finite element (FE) methods and concluded that bodily movement was achieved using attachments. Goto et al. reported that attachments do not influence tensile forces and tipping moment [17]. Many studies have attempted to investigate the effect of attachment shape on aligner behavior. Previous studies have reported that the thickness and shape of the attachment had an effect on tooth movement [18,19,20]. These studies reported that the attachment with a rectangular shape and thickness of 1 mm on the lingual and buccal sides was effective for tooth extrusion and rotation movement [18,19]. Our previous study investigated the optimal shape and position of attachments on the canine by comparing various shapes and positions of attachments for the four movements of tooth: extrusion, intrusion, torque, and rotation [20]. It concluded that the optimal shape of an attachment was a cylinder and should be located lingually rather than buccally to induce effective tooth movement [20].

Although unintended tooth tilting and rotation phenomena were reduced due to the attachments, the moment by the distance between the attachment position and center of the tooth caused the tipping, rotation, and inclination in terms of the mechanism of tooth movement [21,22]. For that reason, more research is needed to address the minimization of the distance between the center of the tooth and the attachments. We devised an overhanging attachment design that can minimize the distance between the center of the tooth and the attachments.

The hypothesis of this study was that overhanging attachment would effectively control tipping, rotation and inclination compared to general attachments. Therefore, the purpose of this study was to derive an effective attachment design that can efficiently induce tooth movement by comparing and analyzing the movement and rotation of teeth between a general attachment and an overhanging attachment.

## 2. Materials and Methods

### 2.1. FE Model Creation

The 3D FE model for the orthodontic treatment was created using mandibular cone-beam computed tomography (CBCT) images used in our previous studies (Figure 1A) [20,23,24]. The CBCT images were imported, segmented into teeth and bone, and converted into 3D models using the commercially available Mimics software (Mimics Research v19.0, Materialise, Leuven, Provincie Vlaams-Brabant, Belgium) (Figure 1B,C). The bone was offset by 2 mm to create the cancellous and cortical bones [23,24,25,26] using SolidWorks (Solidworks 2016, Dassault Systemes SolidWorks Corp., Waltham, MA, USA) (Figure 1B,C). Similarly, the gingiva was added by offsetting the alveolar bone by 2 mm (Figure 1B,C) [20,26]. The periodontal ligament (PDL) was created between the cancellous bone and the teeth as 0.2 mm offsets as reported in the literature [27]. The bones, teeth, PDL, and gingiva were then imported into the Abaqus software package (ABAQUS CAE2016, Dassault systems, Vélizy-Villacoublay, Yvelines, France) for meshing and assignment of material properties. The individual components were meshed with uniform matching four-node linear tetrahedron elements.

### 2.2. Aligner and Attachment Creation

The aligners with 0.5 and 0.75 mm thickness were created to cover the teeth and part of the gingiva. The aligners were designed using Boolean merging, subtraction, and offsetting functions of SolidWorks (Solidworks 2016, Dassault Systemes SolidWorks Corp., Waltham, MA, USA). The created aligners were classified as aligners with no attachment (NA), aligner with general attachment (GA), and aligner with overhanging attachment (OA). The surface of the teeth was used as the base to construct the aligners to mimic the actual thermoforming of an aligner for orthodontic treatment (Figure 2). The composite attachment was also designed in the same manner. The OA was designed to cover the labial midline surface of the crown of the central incisor and the part of the gingiva where the aligner was designed to cover, as shown in Figure 2. The GA was designed to cover only the labial midline region of the crown of the central incisor. In total, six aligner models were created and analyzed in this present study:10.5-mm-thick aligner with the NA20.5-mm-thick aligner with the GA30.5-mm-thick aligner with the OA40.75-mm-thick aligner with the NA50.75-mm-thick aligner with the GA60.75-mm-thick aligner with the OA

### 2.3. Material Properties and Contact Interactions

Linear elastic material properties were assigned to the teeth, cancellous and cortical bones, gingiva, and attachments, as shown below in Table 1. The PDL was assigned hyper-elastic material properties according to the experimental values published by Natali et al. [28]. The aligner material properties were assigned based on the results of the mechanical tests carried out by Seo et al. [11]. Tooth–aligner interactions were assigned frictionless sliding contact whereas tooth-attachment interfaces were “tied”. The attachments–aligner contact interactions were set as frictional surface-to-surface contacts with a 0.2 coefficient of friction. 

### 2.4. Loading and Boundary Conditions

The birth and death FE analysis method described by Zhou et al. [35] was employed in the loading and boundary conditions of this study. The simulation analysis was divided into two processes (Figure 3). In process 1, the aligner treatment planning was performed; in process 2, the actual aligner correction process was realized. In process 1, only the teeth and aligner were analyzed. The teeth were treated as rigid bodies, whereas the aligner was treated as a deformable body. The inside surface of the deformable aligner was “tied” to the rigid teeth and a 0.1 mm displacement in the mesial direction was applied through the center of rotation of the central incisor. The remaining teeth were constrained in all directions of motion. The displaced central incisor, remaining teeth, and the deformed aligner were then exported to process 2. The imported deformed aligner created a mismatch with the deformable teeth of the dental arch. 

To resolve the contact overlaps, process 2 was further divided into two steps. In the first step, the elements of the dental arch comprising of the teeth, cortical and cancellous bones, gingiva and PDL were “deactivated”. The imported rigid teeth and deformed aligner were again “tied,” and a displacement of 0.1 mm in the opposite direction to process 1 was applied to the center of rotation of the central incisor. This generated stresses in the aligner and also resolved the contact mismatch between the aligner and the teeth of the deactivated dental arch. In the second step of process 2, all deactivated elements were “reactivated”. The rigid teeth were deactivated, and the stresses in the aligner were allowed to relax. The relaxation of the stresses in the aligner induced motion in the teeth and thus generated stresses in the PDL, gingiva, and bones. The same processes were followed for the aligners with attachments. Both sides of the alveolar bone were fixed in all directions of movement.

## 3. Results

### 3.1. Tooth Movement

In our study, a 0.1 mm of mesial displacement was applied to analyze the movement of central incisors according to various attachment models.

After the displacement was applied, the movements of the teeth for crown tipping, shaft rotation, and lingual inclination were evaluated (Figure 4).

The highest crown tipping for the 0.5-mm-thick aligner models was measured for the NA (0.394°). The GA tipping was measured to be 0.389°, whereas the OA was measured to be 0.377° (Table 2). The value of the OA was lower than the NA and the GA (3.08% and 4.31%, respectively). For the 0.75-mm-thick aligner models, the GA was measured at 0.391°, whereas the NA and the OA were measured at 0.39° and 0.38°, respectively (Table 2). The value of the OA was lower than the NA and the GA (2.81% and 2.56%, respectively). In both 0.5 mm and 0.75-mm-thick aligner models, the crown tipping values were lowest in the OA.

All models recorded the same buccal inclination of 0.084° for the 0.5-mm-thick aligner models (Table 2). For the 0.75-mm-thick aligner models, the GA recorded the highest inclination (0.085°) while the NA and the OA recorded the same inclination of 0.083° (Table 3).

The NA recorded the highest axial rotation for both the 0.5- mm and 0.75-mm-thick aligner models (0.035° and 0.034°, respectively). GA measured 0.13° for the 0.5-mm-thick aligner and 0.025° for the 0.75-mm-aligner. OA recorded the lowest axial rotation for all models. For the 0.5-mm thick aligner, the value of the OA was lower than the NA and the GA (80% and 46.15%, respectively) (Table 2). In the case of the 0.75-mm-thick aligner, the OA was lower than the NA and the GA (97.6% and 96%, respectively) (Table 3).

### 3.2. Stress Distributions

The peak von Mises stress (PVMS) values and distributions in the aligner and the attachment, as well as the maximum principal stress (MPS) distributions in the cancellous and cortical bones and the PDL are shown in Figure 5 for the 0.5-mm-thick aligner models.

In 0.75-mm-thick aligner models, the peak von Mises stress (PVMS) values and distributions in the aligner and the attachment, as well as the maximum principal stress distributions in the cancellous and cortical bones and the PDL are shown in Figure 6.

The highest MPS values were recorded in the cortical bones for all models. The MPS values of the cortical bone increased with increasing aligner thickness except for OA (Figure 7c). Similarly, the stresses of the cancellous bone, the gingiva and the PDL increased with increasing aligner thickness except for OA (Figure 5, Figure 6 and Figure 7d–f). Higher stresses in the aligner and attachment were recorded in the general attachments than the overhanging attachments for both 0.5-mm and 0.75-mm-thick aligner models (Figure 5, Figure 6 and Figure 7a,b). As shown in Figure 5 and Figure 6, the maximum principal stresses for the PDL for all models were located at the distal midsection.

## 4. Discussion

Despite the growing global demand for clear aligners for the orthodontic treatment of malocclusions, there are still concerns regarding the efficiency of clear aligners to achieve complex tooth-controlling movements. This could be attributed to the lack of clarification on the force/moment-transmission mechanism of aligners [36]. Contrary to the traditional bracket-archwire system, the exact point for force transmission of aligners remains ambiguous. Since tooth movements in the aligner were caused by unintended tilting motions, previous studies have reported the difficulty of achieving the intended tooth movement using aligners [37,38]. 

Composite attachments on tooth surfaces have been demonstrated to result in desirable tooth movements [32,39,40]. Several studies have investigated the effect of the shape and positioning these attachments to have on tooth movement [17,20,29,32,41]. Rossini et al. [37] reported that rectangular or ellipsoid composite attachments improved the quality of orthodontic tooth movements in their systematic review. In this present study, we developed a 3D finite element model of the mandible that is capable of being used to investigate an efficient “overhanging” attachment design that can efficiently induce desired tooth movements. 

The results of our study suggest that for a 0.1 mm distal translation planning for the mandibular central incisor using an aligner, there should be expected crown tipping, axial rotation, and inclination. The aligner force acts on the clinical crown and not through the center of resistance of the tooth, thus producing unintended movements. This means that during treatment with aligners, tooth movements unintended by the orthodontist will occur. When a standard 0.5-mm-thick aligner without any attachment is replaced by a 0.75-mm-thick aligner with the NA, crown tipping, axial rotation, and lingual inclination are not significantly altered. Conversely, the presence of the GA on the tooth surface can reduce axial rotation by 62.86% for a 0.5-mm-thick aligner and 26.47% for a 0.75-mm-thick aligner, respectively. For both aligner thicknesses, crown tipping and lingual inclination did not vary significantly. This observation is consistent with Ho et al. [42], who reported that their attachments did not prevent tipping. Compared to an aligner with the NA, an aligner with the OA will reduce axial rotation by 80% and 97.06% for a 0.5 mm- and 0.75-mm-thick aligner, respectively. The efficiency of the overhanging aligner to reduce axial rotation can be ascribed to the increased area of application of the aligner orthodontic force to the tooth. While crown tipping was reduced by 4.31% and 2.56% by a 0.5 mm- and 0.75-mm-thick aligner with the OA, respectively, lingual inclination remained similar to the aligner with the NA. This suggests that an effective attachment is not able to completely prevent lingual inclination but can reduce axial rotation and crown tipping.

The stress distribution patterns in the cancellous bone, cortical bone, gingiva, and PDL were similar. The MPS values for the cortical bone, cancellous bone, and gingiva were not significantly different in both 0.5-mm- and 0.75-mm-thick aligner models, indicating a similar bone response. The residual aligner PVMS in the GA models were the highest for both aligner thickness models. The OA residual aligner PVMS was about 1.7 times more than NA, whereas the GA residual aligner PVMS was about four times more than NA. It can thus be concluded that the presence of attachments on the surface of the tooth will induce more stress in the aligner, which leads to more tooth movement. 

Extremely high PVMS values were measured for the GA. Thus, we predicted that for the aligner with the GA, there would be a high risk of yield of the attachment and a high possibility of detachment of the attachment. The highest stress in the incisor PDL was concentrated at the middle part in the distal region. Tensile stresses were mainly recorded on the distal side while compressive stresses were located on the mesial mid-region. The stress values measured for the PDL in all models were within 97–106 MPa in tension and 98–106 MPa in compression. These values are within the magnitude of stress that can alter the PDL for the onset of bone remodeling [43,44].

Our study did not account for the effect of the masseter muscle while wearing the aligner and other masticatory movements. The effect of the masseter muscle and masticatory movements must be considered as the masseter muscle has been reported to influence craniofacial characteristics, malocclusions and asymmetry [45]. The OA and the GA were compared by simply moving the tooth in the lateral direction. However, in an actual orthodontic treatment situation, since the incisor teeth were in an oblique position, it is necessary to analyze the rotational moment of the root to compare the GA and the OA. In addition, research is needed for various tooth movements such as extrusion, intrusion, rotation, etc. Therefore, future studies considering the stress distribution and movement characteristics between the GA and the OA should compare and analyze various tooth movements such as extrusion, intrusion, rotation, and torque. Again, it is recommended to conduct future research by considering the movement of the root of the teeth and other complex teeth movements.

## 5. Conclusions

Based on the results of our study, we confirmed that the OA can control the orthodontist’s unintentional tooth movement better than the GA. The OA is considered to reduce the risk of detachment of the attachment during orthodontic treatment by showing desirable stress distributions and reducing the stress concentration between the attachment and the aligner. Therefore, the OA is an effective attachment design on the surface of the tooth that can efficiently induce bodily tooth movement with minimal unintended axial rotation and crown-tipping tooth movements.

## Figures and Tables

**Figure 1 materials-14-04926-f001:**
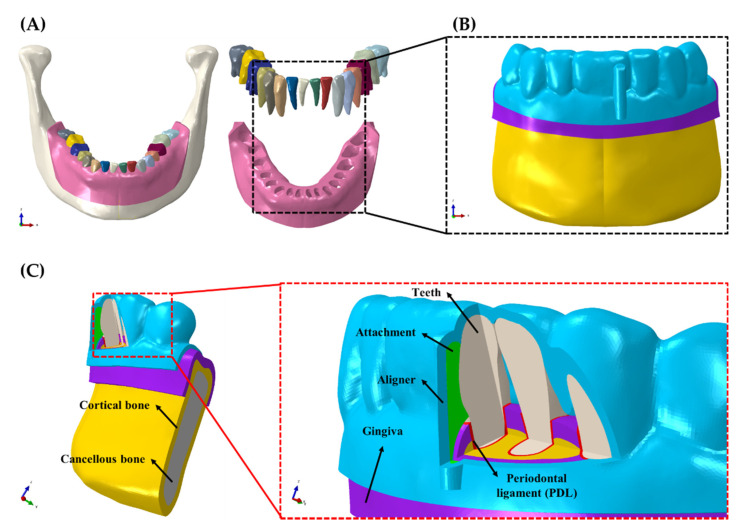
A 3D FE creation process in the orthodontic treatment situation. (**A**) Bone, gingiva and teeth segmentation from 3D CBCT data; (**B**) 3D orthodontic FE model. (**C**) Component of 3D orthodontic FE model.

**Figure 2 materials-14-04926-f002:**
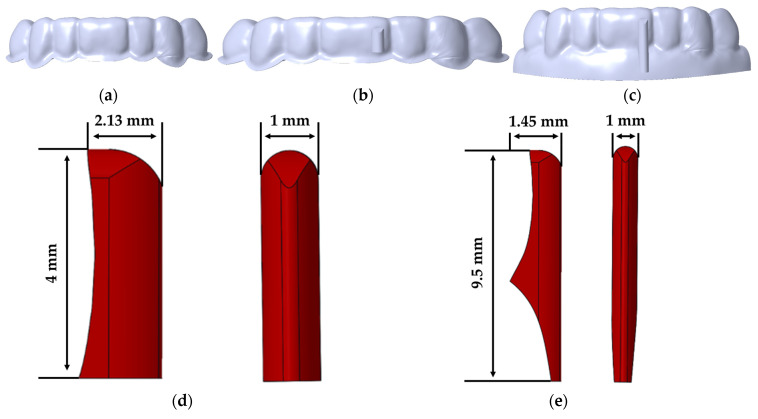
A 3D aligner model design. (**a**) Aligner with the NA; (**b**) aligner with the GA; (**c**) aligner with the OA; (**d**) shapes and dimensions of the general composite attachment; (**e**) shapes and dimensions of the overhanging composite attachment.

**Figure 3 materials-14-04926-f003:**
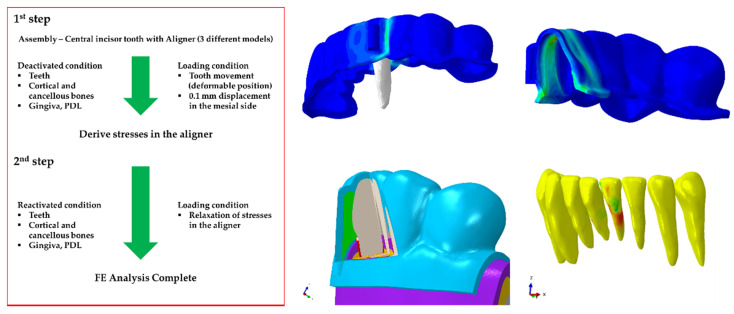
FE analysis process of three different orthodontic treatment models (NA, GA, OA).

**Figure 4 materials-14-04926-f004:**
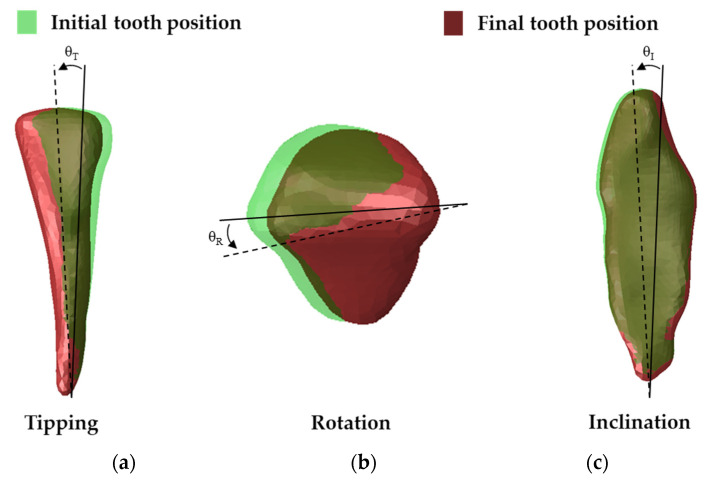
Tooth movements before and after loading conditions. (**a**) tipping; (**b**) axial rotation; (**c**) buccolingual inclination.

**Figure 5 materials-14-04926-f005:**
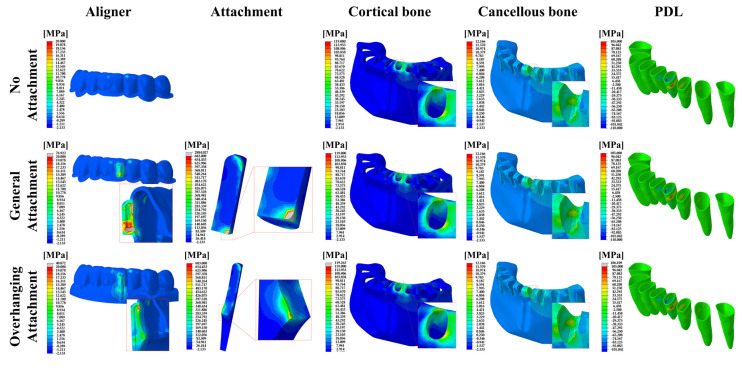
FE results of the stress distributions for the components of the orthodontic treatment model in the 0.5 mm aligner models. The aligner and attachment were expressed as PVMS values, and the bones (cortical and cancellous) and PDL were expressed as MPS values.

**Figure 6 materials-14-04926-f006:**
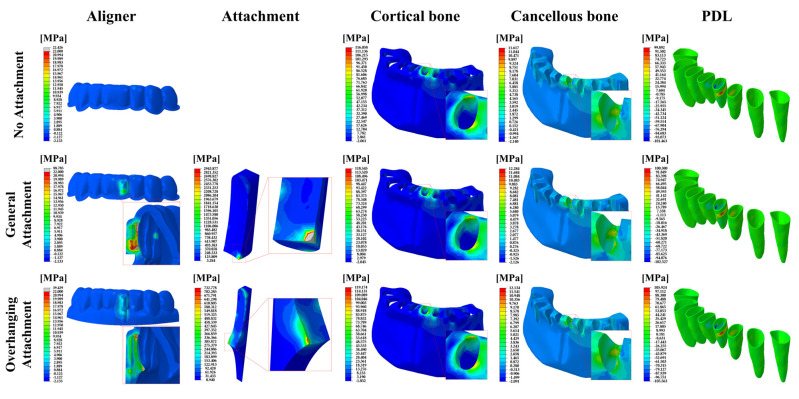
FE results of the stress distribution for the component of the orthodontic treatment model in the 0.75 mm aligner models. The aligner and attachment were expressed as PVMS values, and the bones (cortical and cancellous) and PDL were expressed as MPS values.

**Figure 7 materials-14-04926-f007:**
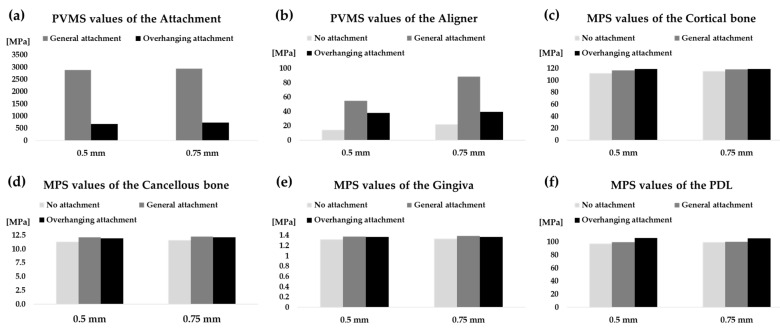
PVMS values measured in the aligner and attachment, and MPS values measured in the cortical bone, cancellous bone, gingiva, and PDL for both 0.5-mm and 0.75-mm-thick aligner models. (**a**) Attachment; (**b**) aligner; (**c**) cortical bone; (**d**) cancellous bone; (**e**) gingiva; (**f**) PDL.

**Table 1 materials-14-04926-t001:** Material properties of the linear components of the FE model.

Component	Elastic Modulus [MPa]	Poisson’s Ratio	Reference
Cancellous bone	1370	0.3	[20,26,29]
Cortical bone	13,700	0.3	[20,26,29]
Gingiva	2.8	0.4	[20,30,31]
Attachment	12,500	0.36	[32]
Teeth	19,613	0.15	[25,33,34]

**Table 2 materials-14-04926-t002:** Tooth movements measured for 0.5-mm-thick aligner models.

Model	Tipping (°)	Rotation (°)	Inclination (°)
No attachment	0.394	0.035	0.084
General attachment	0.389	0.013	0.084
Overhanging attachment	0.377	0.007	0.084

**Table 3 materials-14-04926-t003:** Tooth movements measured for 0.75-mm-thick aligner models.

Model	Tipping (°)	Rotation (°)	Inclination (°)
No attachment	0.390	0.034	0.083
General attachment	0.391	0.025	0.085
Overhanging attachment	0.380	0.001	0.083

## Data Availability

The data presented in this study are available on request from the corresponding author.

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
