# Peer review of "Efficient Design of a Clear Aligner Attachment to Induce Bodily Tooth Movement in Orthodontic Treatment Using Finite Element Analysis"

_materials, 2021, doi:10.3390/ma14174926_

Round 1

Reviewer 1 Report

  1. Line 73 : These CBCT images were deemed to have not had had any history of orthodontic treatment or periodontal diseases. => Is there any mistake ?
  2. How about the effects of age and sex on study model ?
  3. Please to introduce the sizes of stand attachment and overhanging attachment in text.
  4. And, how to determine the amount of gingival part ? 
  5. How to clean overhanging attachment during clinical application in future?
  6. Does the little difference have clinical significance ?

Reviewer 2 Report

General comment:

The present manuscript is interesting and new as a study! Through the work, I have observed some errors and needs to be reformulated in some places; but after the review, this work can be further evaluated for publication. Please find critical comments as follows:

Critical comments:

Abstract

Line 19: Put “adult” before patients. Because the clear aligner system is mostly for the adults and you have to specify this, and not in every malocclusion it can be used.

Line 25: Don’t start the sentence with a number…here you can say “The 3D…”.

Line 26: Delete “mm” after 0.5, it is not necessary.

Lines 26-27: Reformulate the sentence…”with general and overhanging one”.

Keywords: I suggest to add “treatment or therapy” after orthodontics; what you prefer mostly.

  1. Introduction

Please enrich this section!

Line 43: Here you can mention the role of clear aligners for a better oral hygiene; this system avoids also the apical resorption, as a consequence the latter… I suggest to the authors this article that is interesting and can help to improve the introduction: [Efficacy of a Copper-Calcium-Hydroxide Solution in Reducing Microbial Plaque on Orthodontic Clear Aligners: A Case Report. Eur J Dent. 2019 Jul;13(3):478-484. doi: 10.1055/s-0039-1695653.]

Line 50: I suggest to put into parentheses (FE) after the long name and deleting “M”. In this manner you can maintain the acronym FE for the entire manuscript.

  1. Materials and Methods

This section has a lot of information and I appreciate this, but also here are some points to be corrected.

Line 70: Here I think is a mistake in the title…before D is something else? I suggest to put …FE model creation…

Line 87: Put “A” before 3D.

Line 88: You can change the point (B) in this way “3D orthodontic FE model”.

Line 89: I think that the whole parentheses is unnecessary, you can delete it, because is clear the figure.  

Lines 91: Please improve the sentence, because in English grammar it is not correct to start the new sentence with a number.

Line 98: Put OA and not the long name.

Line 100: Put GA and not the long name.

Lines 103-108: Put their respective acronyms for long names, because you used them in lines 94-95.

Line 109: Again here, put “A” before 3D. Also replace the long names with their respective acronyms.

Line 122: What is FEA? If you have mention this before as a long name, please put the acronym near its long name.

Line 145: Put the respective acronyms in parentheses and not the long names, while for “long attachment” put next to its acronym (LA).

  1. Results

Lines 149-151: It is very long as a sentence and not very clear. Please, reformulate it!

  1. Discussion

This section needs improvement. You need to mention your newest of this work!

Lines 207-209: Not very clear as a phrase.

- Please, replace the long names with their respective acronyms… FE, OA, GA, NA.

Line 261: Please, improve the sentence, because “In addition, additional…” is repetitive.

  1. Conclusions

This section requires to be clearer! Replace the respective acronyms.

Line 269: Here you can improve in English the beginning of the sentence.

Lines 273-275: I think isn’t necessary this sentence. Maybe you can delete and maintain the last sentence.

Reviewer 3 Report

Efficient Design of a Clear Aligner Attachment to Induce 2 Bodily Tooth Movement in Orthodontic Treatment using Finite 3 Element Analysis is a very interesting paper that propose effective an attachment design that can efficiently induce tooth  movement. In my opinion some correction are necessary before it can be considered valid for pubblication. 

INTRODUCTION

The introductory part is too generic, the importance
of attachments'
geometry for the determination of the different
dental moovements with clear aligners should be described
in a more extensive way.

MATERIALS AND METODS

this section is well structured and clearly describes the study design, the assessed outcomes and the statistical methodology used correctly for the interpretation of the data

RESULTS

the data obtained in the study are clearly reported and the
representations in the table are well formulated. Several hard
and soft tissues that are involved in orthodontic tooth movement
were considered in the study outcome assessment, however the
forces related to the masticatory muscles were not considered
and this could represent a bias for the study.

DISCUSSION
overall well-articulated, comprehensive comparison with other
studies in the literature, description of the limits of the
research and the meaning of the results
